# Innovative Quality Assessment of Pavement Subgrades Using the Glegg Impact Soil Tester

**Katarina Hodasova** [1,*], **Juraj Musuta** [2], **Martin Decky** [1] **and Maria Kudelcikova** [3]

1   Department of Highway and Environmental Engineering, Faculty of Civil Engineering, University of Zilina, Univerzitna 8215/1, 010 26 Zilina, Slovakia; martin.decky@uniza.sk
2   Department of Construction Management, Faculty of Civil Engineering, University of Zilina, Univerzitna 8215/1, 010 26 Zilina, Slovakia; juraj.musuta@uniza.sk
3   Department of Structural Mechanics and Applied Mathematics, Faculty of Civil Engineering, University of Zilina, Univerzitna 8215/1, 010 26 Zilina, Slovakia; maria.kudelcikova@uniza.sk
*   Correspondence: katarina.hodasova@uniza.sk

**Abstract:** This article presents the case study of our research in the field of innovative methods of pavement subgrade quality control using the CIST (Clegg Impact Soil Tester) device. The CIST device developed by Dr Clegg from the University of Western Australia measures soil compaction indirectly using the CBR value. The value is evaluated based on the deceleration rate of a falling 4.5 kg weight moving in a vertical guide roller. In Europe, for the assessment of the mechanical efficiency (bearing capacity) of cohesive soils in the pavement subgrade, priority is given to indirect assessment methods especially using the laboratory determination of CBR (Californian Bearing Ratio) and directly through the implementation of a static plate load test (SPLT). This article reports the long-term results of our research in the field of verification and validation of an innovative CIST device, which minimizes the time, space, and economic disadvantages of SPLT. This article presents the results of determining the field of applicability of the CIST device for cohesive soils, the correlation dependencies (CD) of the CBR values determined by the CIST device, and, according to STN 72 1016, the CD of the impact dynamic deformation modulus $E_{vd}$ from the CIV (Clegg Impact Value). We consider the most important results of our long-term research to be a recognition of the ability of CIST to assess the quality of cohesive soils up to a compression value of 40 mm, corresponding to a CBR of 2.2% and a modulus of subgrade deformation of 20 MPa. A very strong correlation dependence of $CBR_{Clegg}$ [%] on the moisture content of clayey soils in the interval from 5 to 19% was also observed. The presented knowledge led to the creation of relevant documents for the credible implementation of the CIST device in the system approach for assessing the quality of the pavement subgrade.

**Keywords:** subgrade; pavement; Clegg Impact Soil Tester; CIST; quality control; LDD 100

## 1. Introduction

The insights presented in this article are convergent with the objective of Horizon Europe, approved by the European Parliament on 17 April 2019. Thus, it has become the European Union's framework program for research and innovation for 2021–2027. It represents the EU's major initiative to support research and innovation from conception to market application. It complements the national and regional funding. The structure of the proposed Horizon Europe program consists of three pillars: excellent science, global challenges, and competitiveness of the European industry and innovation. A complementary cross-cutting part of the program would introduce measures to enable the EU to exploit its full potential in research and innovation.

In the field of road engineering, Slovakia has significant potential for innovation in a university environment. Consequently, in recent times, the employees of the Faculty of Civil Engineering of UNIZA have published articles on the development of innovative

building materials [1] and their quality control [2]. Their research activities were based on the perception of innovation, for which credible insights are provided in [3].

Schumpeter, who can be considered the founder of the theory of innovation in economics, defined innovation as the assertion (i.e., not just invention) of a new combination of factors of production (by entrepreneurs) [4]. Innovation involves creating a new idea and subsequently implementing it in a new product, service, or process. It leads to dynamic economic growth and increased employment and generates a net profit for the innovative enterprise. Innovation is never a one-off phenomenon. It is a long and cumulative process involving many organizational decision-making processes, from the phase of generating a new idea to the phase of its implementation. A new idea relates to the perception of a new customer need or a new production method. It is developed in a cumulative process of gathering information associated with a constantly challenging business vision. The new idea is developed and commercialized into a new marketable product or process through the implementation process, accompanied by cost reductions and productivity improvements [5]. According to [6], innovation is a process that combines science, technology, economics, and management. It is meant to achieve novelty and ranges from the origin of an idea to its commercialization in the form of production, exchange, and consumption.

In this case study, the authors present the research results on innovative processes for assessing the quality of compaction of unbound structures (earth structures) of transportation structures. In Slovakia, decisive methods for quality control of earth structures are carried out in terms of STN 73 6190 [7] and STN 73 6192 [8]. The normative process of quality assessment by static plate load testing has significant methodological, spatial, and time limits, and the resulting economic demands. In the case of small-scale traffic structures, the innovative methods discussed in this article can be an effective tool in this area. For assessing the compaction quality of fine-grained cohesive soils in the subgrade, the Clegg Impact Soil Tester (CIST) is an alternative. The presented recommendations can be applied from the aspect of the provisions of the Road Act [9]. The Act states that the road design and construction are carried out in accordance with the applicable Slovak technical standards, technical regulations, or objectively determined results of research and development for road infrastructure, or similar technical specifications.

## 2. Infrastructure Used to Identify Interest Correlations

This article describes the results of the author's research on the implementation of the CIST device in a system approach for pavement subgrade quality assessment. Dr Baden Clegg developed the concept of the CIST device [10] in the late 1960s while lecturing at the University of Western Australia in Nedlands (Perth). The University of Western Australia established a marketing division in 1976 to promote the device and make research results available for the quality control of earth construction. In honor of Dr Clegg, the CIST output parameter was named the Clegg Impact Value (CIV). In 1993, Dr. Clegg founded the company bearing his name. The aim was to provide a worldwide information service supporting research and development cooperation in the quality control of earth structures through CIST [11]. Since 2005, the authors considered the possibility of using CIST type WS 32830 (Figure 1) for quality control of fine-grained soils and bulk materials [12].

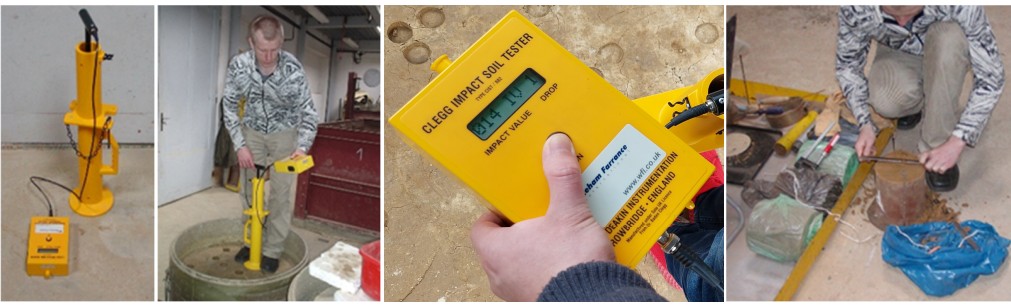

**Figure 1.** Device CIST-Type 31820 and homomorphic clay pavement subgrade models.

This device measures the compaction of soils indirectly using the CBR value. The value is evaluated based on the rate of deceleration of a falling 4.5 kg weight moving in a vertical guide roller. After being released from a certain height, the weight falls in the roller and hits the surface of the foundation slab. The rate of deceleration is determined by the force dependent on the compaction of the material at the point of compaction. The device provides practically immediate results for the rate of compaction of the soils under consideration. This eliminates the major disadvantages of other methods of compaction quality control (determination of bulk density, static load tests, and geodetic control methods). The operation of the device is simple, and the handling of the device is physically easy due to its weight.

All measurements with the CIST device were carried out strictly according to the instructions for the use of the CIST device type WS 32830 (Figure 1). The CIV measurement was always recorded after the fourth fall of the hammer when the fifth measurement met the required deviation.

As part of the research activities, measurements were carried out primarily on the following models:

- Homomorphic clay subgrade models (Figure 1);
- Isomorphic pavement model built within the Scientific Research Workplace of the Faculty of Civil Engineering of the University of Žilina (Figure 2);
- Experimental field for research on unbonded pavement structures (Figure 3).

The next device was a lightweight dynamic plate LDD 100 representing a falling weight deflectometer (FWD) with a light weight (Figure 4).

According to the Slovak technical standard STN 73 6192, the basic formula for calculating the modulus of dynamic deformation $E_{vd}$ is as follows:

$$E_{vd} = \frac{\pi}{2} \cdot \left(1 - v^2\right) \cdot \frac{a \cdot \sigma}{y_{m1}} = \frac{\pi \cdot d \cdot \sigma}{4 \cdot y_{m1}} \cdot \left(1 - v^2\right) \tag{1}$$

The variables in Formula (1) are as follows: $a$ radius of the loading plate [m], $d$ diameter of the loading plate [m], $\sigma$ contact stress [MPa], $y_{m1}$ amplitude of deflection at the center of the loading plate [m], and $v$ Poisson's ratio [-].

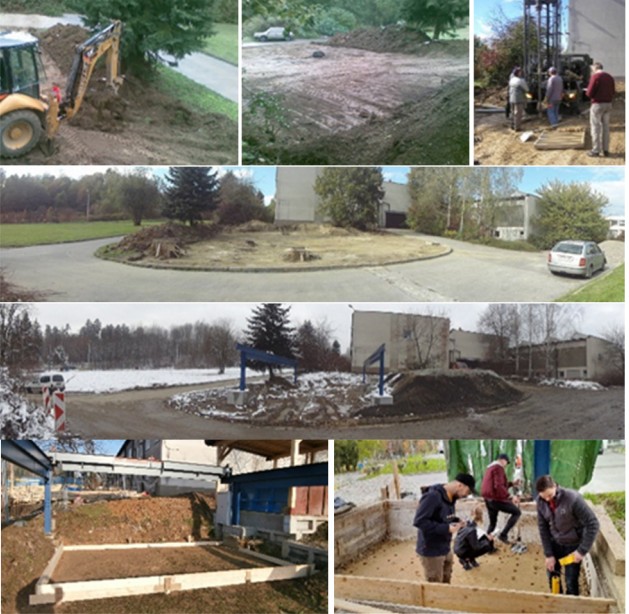

**Figure 2.** Construction of the Scientific Research Workplace of the Faculty of Civil Engineering at the University of Žilina and measurements of the quality of the clay subgrade.

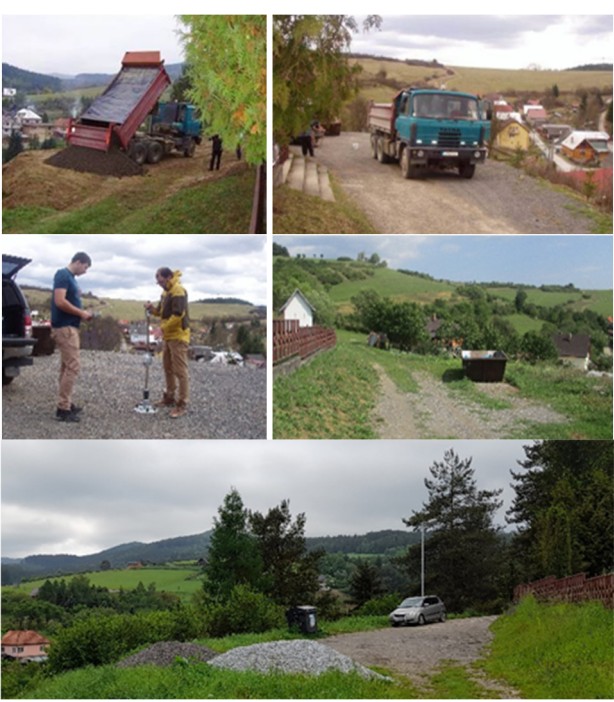

**Figure 3.** Construction of the experimental field for research on unbonded pavement structures.

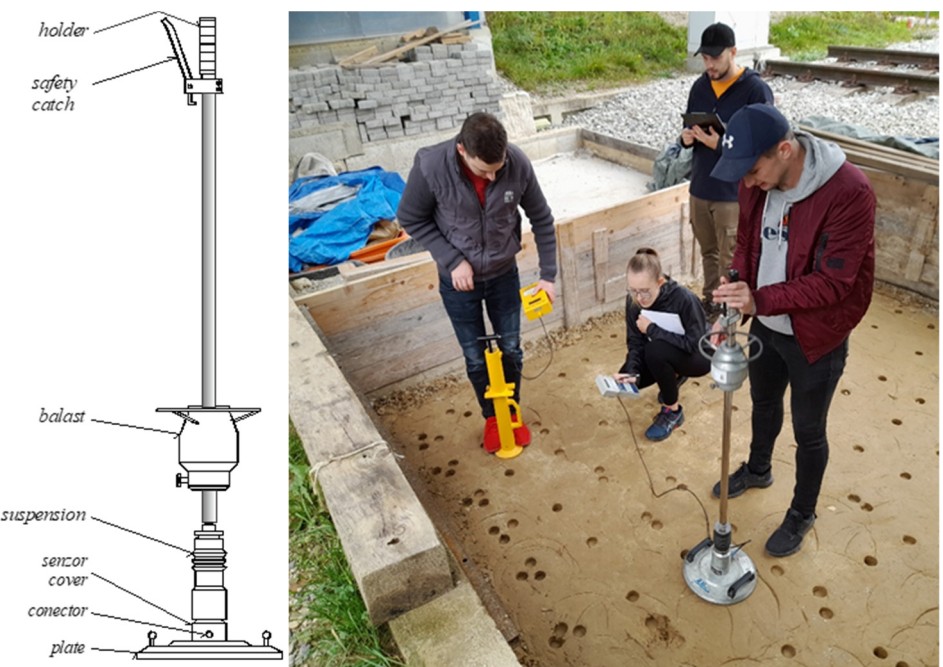

**Figure 4.** Schematic of LDD 100 and views of CIST and LDD 100 measurements at the Scientific Research Workplace SvF UNIZA.

## 3. Verification of the Suitability of CIST Device for Quality Control of Cohesive Soils

The CIST device, according to the original methodology [11], can be used to evaluate the compaction of cohesive soils for penetrations of less than 20 mm. To verify this condition and the hypothesis that the device measures CBR values identical to or correlated with those determined according to STN 72 1016 [13], laboratory measurements were carried out on the following soils: sandy clay and clayey gravel. Measurements were performed on large-scale rings with clayey soils and on the soil samples used for the CBR test (Figure 1) according to STN 72 1016. The moisture content of the clayey soil test samples ranged from

4 to 19%. For the moisture range indicated, the clayey soil samples reached a bulk density level of 1710 to 1960 kg·m$^{-3}$. Measurements were taken for each sample on both the upper and lower surfaces each time.

Figure 5 illustrates the correlation dependence of the Clegg hammer push $y_{Clegg}$ [mm] into the clay soil samples (Figure 1) measured after its fourth fall from the CIST device and from the Clegg impact value (CIV) read from the device after the fourth fall. Figure 5 shows that a compaction of 20 mm corresponds to a CIV value of 7. According to the equipment manual, the fourth compaction value can be converted into a CBR equivalent using the following formula:

$$CBR_{Clegg} = (0.24 \cdot CIV + 1)^2 \tag{2}$$

where $CBR_{Clegg}$ is the CBR value [%] evaluated according to Formula (2). For the identified threshold value of CIV = 7, we obtained the following value:

$$CBR_{Clegg} = (0.24 \cdot CIV + 1)^2 = (0.24 \cdot 7 + 1)^2$$

$$CBR_{Clegg} \doteq 7.2\%$$

The dependencies of the 14 values of the correlated variables were evaluated using several forms of correlation dependencies (CD): linear, power, exponential, and polynomial. The linear (R = 0.9819) and exponential (R = 0.9890) correlation coefficients demonstrated the highest correlation coefficient value. Based on the logical premise that CIV values can take only positive values, it was possible to consider the exponential correlation dependence only. When assessing the quality of earth structures made of fine-grained soils (clays, sands, loamy soils, etc.), we can use the measured CIV value [-] to calculate the compression $y_{CIST}$ [mm] according to Formula (3).

$$y_{CIST} = 53.717 \cdot e^{-0.148 \cdot CIV} \tag{3}$$

From Formula (3), we can determine the value of CIV from the measured value of $y_{CIST}$ [mm] according to Formula (4).

$$CIV = -\frac{ln(y_{CIST}/53.717)}{0.148} \tag{4}$$

Another verified premise was to validate the ability of the CIST to accurately detect the effect of clay soil moisture on their CBR values. The laboratory-observed dependencies of the Clegg values on the moisture content of the tested samples are presented in Figure 6.

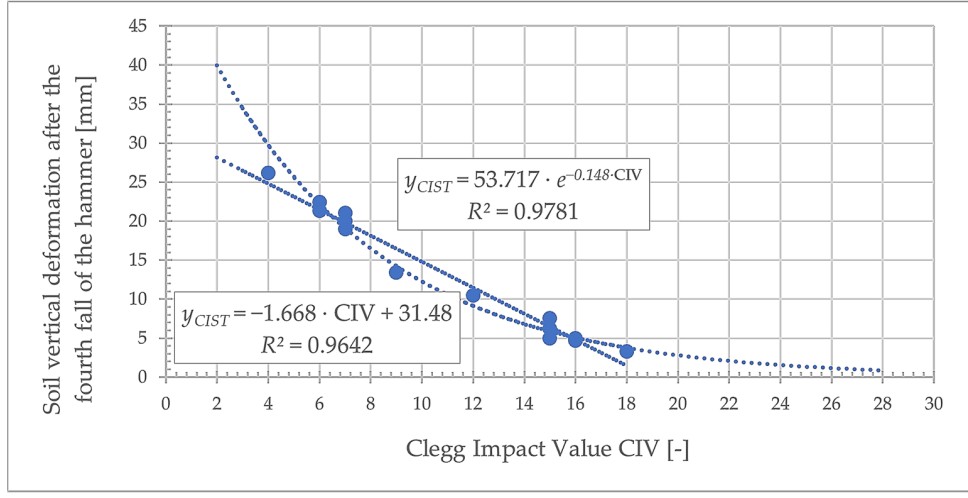

**Figure 5.** Laboratory-determined dependence of the CIV hammer push into the samples of clay and loam subgrade of pavements.

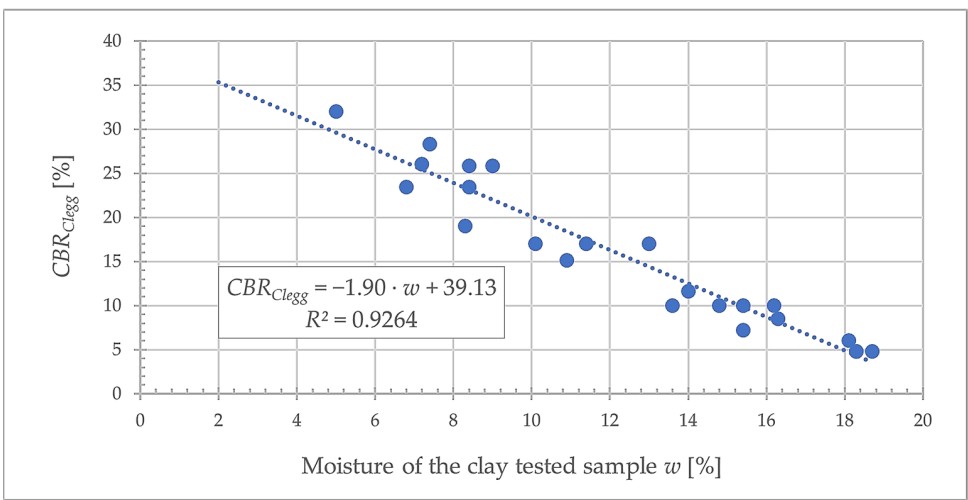

**Figure 6.** Laboratory-objectified dependence of Clegg CBR values on clay soil moisture.

A credible assessment of objectified correlation dependencies necessitates an overview of the most commonly utilized correlation characteristics. In a broader sense, correlation is any statistical relationship between two random variables in bivariate data. It is crucial to note that correlation does not always imply causation. The degree of causal dependency is expressed through the coefficient of determination, which is a fundamental output in the regression analysis.

The correlation coefficient is a statistical measure gauging the intensity of the relationship between the relative movements of the two variables. Its values range from $-1$ to $+1$. In practice, Pearson's correlation coefficients and Spearman's rank correlation coefficients are the most commonly used.

Pearson's correlation coefficient is a statistic that measures the linear relationship between a pair of random variables (*X*,*Y*). The formula for the Pearson correlation coefficient $R(X,Y)$ involves the covariance $cov(X,Y)$ and standard deviations $\sigma_X$ and $\sigma_Y$, which can be expressed as follows:

$$R(X,Y) = \frac{cov(X,Y)}{\sigma_X \sigma_Y} = \frac{\sum_{i=1}^{n} X_i Y_i - n\overline{XY}}{\sqrt{\left(\sum_{i=1}^{n} X_i^2 - n\overline{X}^2\right)\left(\sum_{i=1}^{n} Y_i^2 - n\overline{Y}^2\right)}} \tag{5}$$

where $n$ is the size of the sample, $X_i$; $Y_i$ represents the individual sample points; $\overline{X}$ and $\overline{Y}$ are the sample means. An issue with the Pearson correlation lies in its sensitivity to outliers, potentially resulting in erroneous conclusions depending on the data. Optimal application of the Pearson correlation coefficient requires adherence to specific criteria: the variables are quantitative and normally distributed, the data have no outliers, and the relationship is linear [14].

The coefficient of determination, symbolized as $R^2$, stands as the square of the correlation coefficient. Its range consistently falls between 0 and 1, frequently presented as a percentage. It provides insights into how well the regression model fits the observed data, with a higher $R^2$ indicating more variability. When dealing with small samples, the coefficient of determination is often more reliable than the correlation coefficient [15].

The Spearman rank correlation coefficient is a nonparametric measure of rank correlation, assessing the statistical dependence between the rankings of two variables. Unlike the Pearson correlation, Spearman focuses on monotonic relationships, whether linear or not, making it suitable for variables that are ordinal, not normally distributed, include outliers, or exhibit a non-linear but monotonic relationship. A perfect Spearman correlation of $+1$ or $-1$ materializes when each variable constitutes a purely monotonic function of the other; the formula is as follows:

$$\rho = 1 - \frac{6\sum d_i^2}{n(n^2 - 1)} \tag{6}$$

where $d_i$ is the difference between the *X*-variable rank and the *Y*-variable rank for each pair of data, and *n* is the size of the sample [16]. The Spearman correlation coefficient proves to be a superior option when at least one of the following conditions applies: the variables are ordinal, do not adhere to a normal distribution, outliers are present in the data, or the relationship between variables is not linear but exhibits monotonic patterns [14].

Table 1 shows the range of correlation coefficient values (rounded to two decimals) proposed in the literature [17] for the level of correlation.

**Table 1.** Range of correlation coefficient values and the corresponding levels of correlation.

| Range of Correlation Coefficient Values (*R*, $\rho$) | | Level of Correlation (Positive or Negative) |
|---|---|---|
| 0.80 to 1.00 | −1.00 to −0.80 | very strong correlation |
| 0.60 to 0.79 | −0.79 to −0.60 | strong correlation |
| 0.40 to 0.59 | −0.59 to −0.40 | moderate correlation |
| 0.20 to 0.39 | −0.39 to −0.20 | weak correlation |
| 0.01 to 0.19 | −0.19 to −0.01 | very weak correlation |
| | 0.00 | no correlation |

According to the presented facts, we can conclude that there is a very strong (*R* = 0.9625) correlation dependence (Figure 6) of the $CBR_{Clegg}$ values [%] on the moisture of fine-grained soils *w* [%].

$$CBR_{Clegg} = -1.90 \cdot w + 39.13 \qquad (7)$$

Another hypothesis was to verify whether the CIST device provides a compatible output with the determination of the CBR value according to STN 72 1016 [13]. Since STN 72 1016 yields lower values than the Clegg device CBR determination, correlation relationships of STN 72 1016 CBR values to Clegg CBR values were worked out (Figure 7).

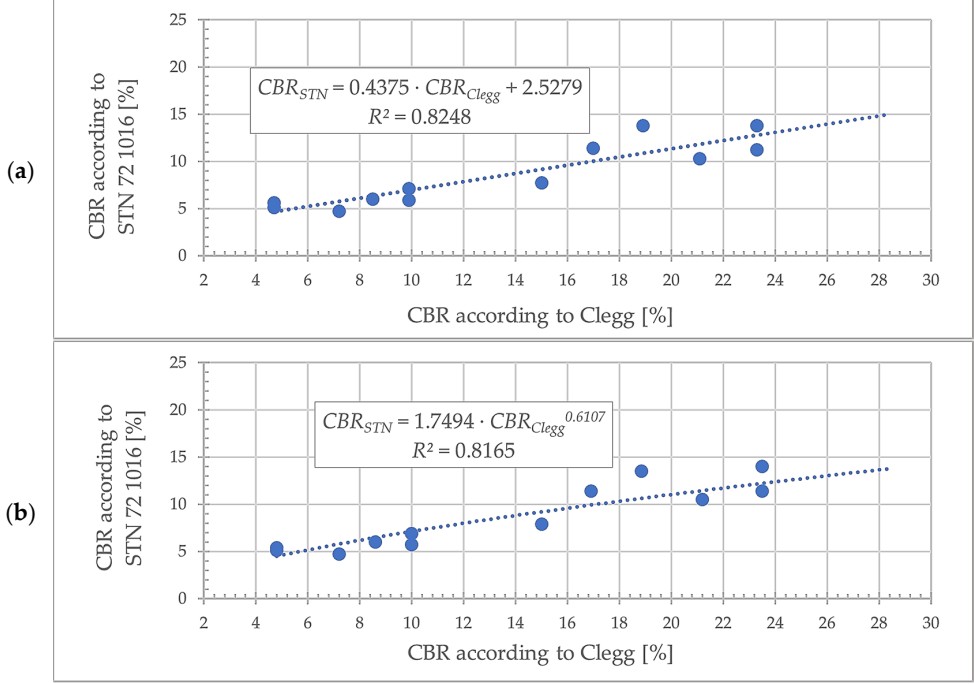

**Figure 7.** Laboratory-determined: (**a**) linear and (**b**) power dependence of CBR according to STN 72 1016 from $CBR_{Clegg}$ for sandy clay.

From Figure 7, it is evident that the $CBR_{Clegg}$ values are significantly different from the $CBR_{STN721016}$ values. On average, they are 1.8-fold higher. Therefore, the $CBR_{Clegg}$ values cannot be used directly in the pavement construction design.

## 4. The Use of CIST Devise for Quality Control of Clay-Soil Pavement Subgrade Construction

In Slovakia, according to the requirements of the Road Act [9], the process of design and construction of asphalt pavements is codified in STN 73 6114 [18] and in the technical conditions TP 033 [19]. According to STN 73 6114, the pavement should be designed to resist, with the required level of reliability and the loads and impacts that can be expected to occur during its operation. The design of the pavement is based on the traffic significance of the road, the traffic load on the road and the climatic conditions, the technological possibilities, the possibilities of utilizing local materials, and the protection of health and the environment.

The asphalt pavement is the paved part of the road intended for vehicle traffic and is composed of the following (Figure 8):

- Asphalt surface;
- Pavement subbase;
- Capping layer.

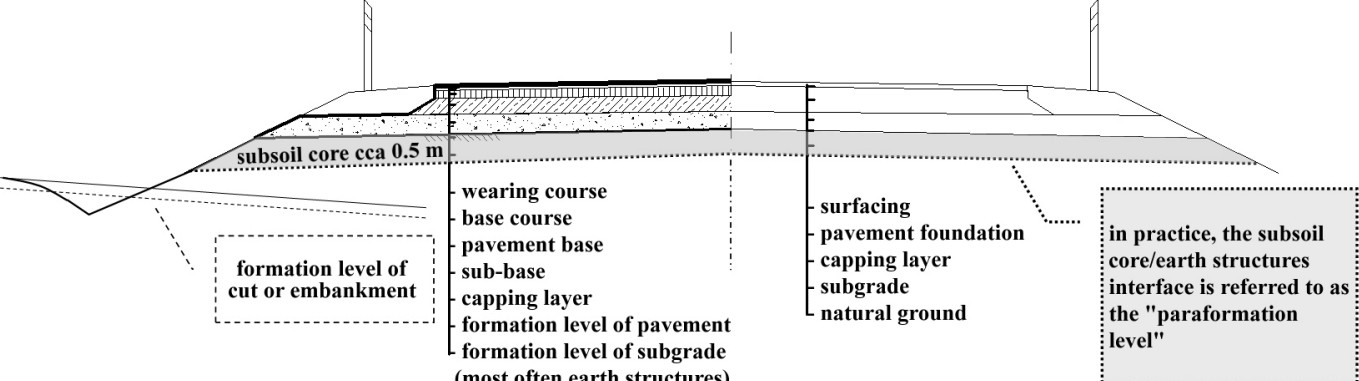

**Figure 8.** Terminology of asphalt pavement construction layers according to the World Road Association (PIARC—Permanent International Association of Road Congresses).

According to the provisions of these regulations, it is necessary to use Figure 9 to convert the CBR to the design value of the elastic modulus of the formation level of the pavement.

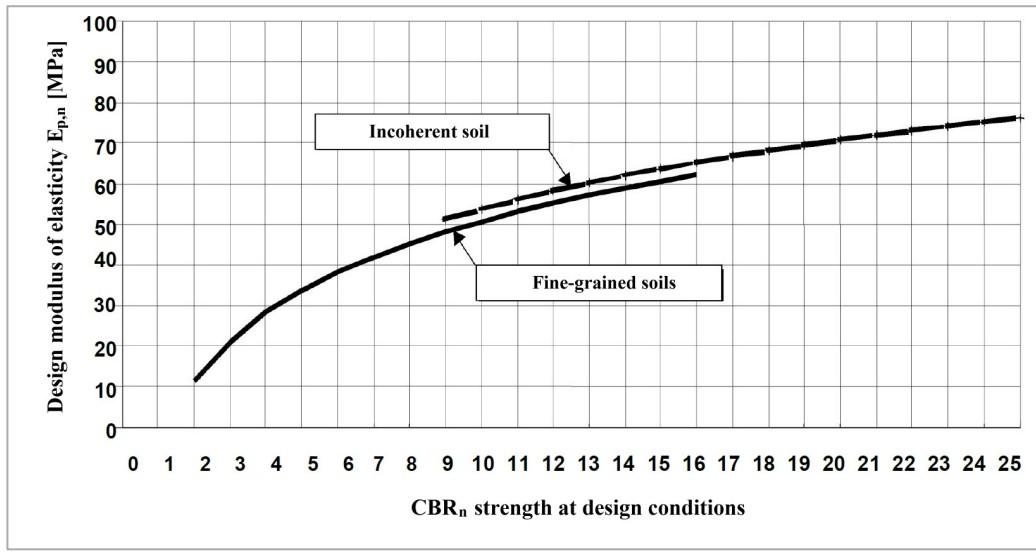

**Figure 9.** Dependence of the modulus of elasticity of the soil on the CBR values according to TP 033 [19].

Since the direct equipollence of the $CBR_{Clegg}$ and $CBR_{STN721016}$ values was not confirmed, it was necessary to consider the possibility of indirect implementation of the CIST outputs into the dimensioning and quality control system for the construction of the unbonded pavement structural layers. For this purpose, research on the correlation dependencies of CBR values [%] from CIV values was carried out. The results are presented in Figure 10.

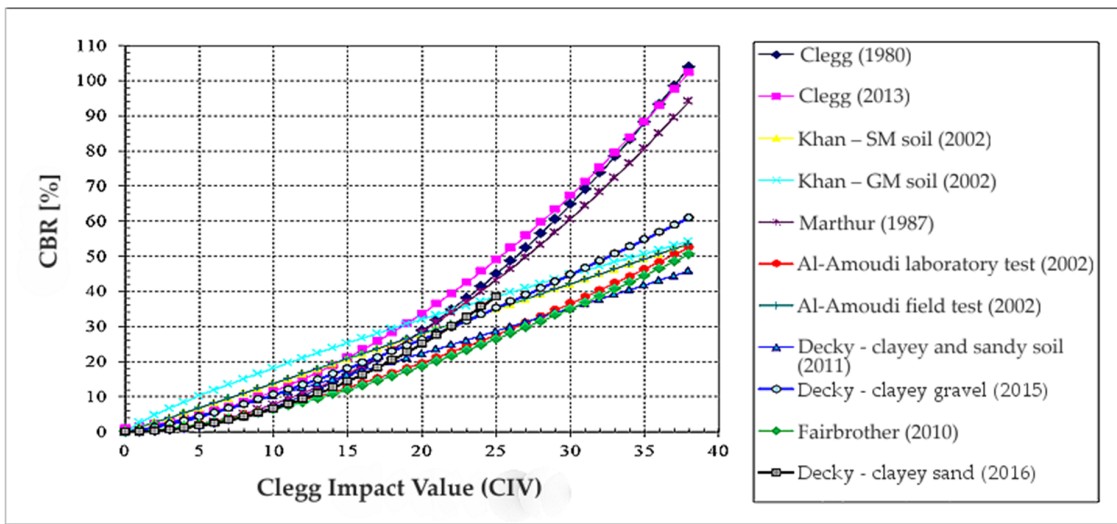

**Figure 10.** Comparn of correlation dependencies of CBR values from CIV.

Figure 10 demonstrates that the calculation values of $CBR_{Clegg}$ according to the original 1980 methodology (slightly modified in 2013) are 1.8-fold higher, on average, than the findings of the other authors [20,21]. The article stated that this fold difference was also objectified by our research.

Measurements of the mechanical characteristics on the surface of unbonded structural layers of the isomorphic pavement model (Figure 11 were carried out within the dissertation thesis of Ing. Katarina Hodasova and Ing. Juraj Musuta. The following devices were used in the evaluation of the characteristics:

- Lightweight dynamic plate LDD 100 (Figure 11)—the device automatically evaluates the dynamic modulus of deformation $E_{vd}$ (the value of the Poisson number is input);
- Clegg Impact Soil Tester–CIST (Figure 11, right)—the output is the CIV (Clegg Impact Value) measured after the fourth fall of the impact hammer.

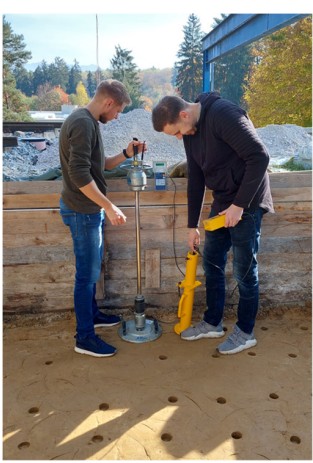 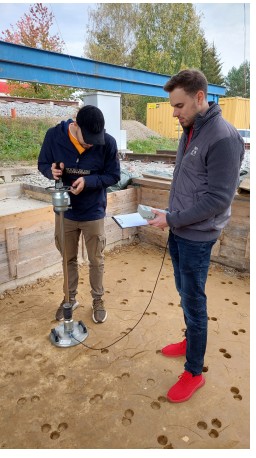 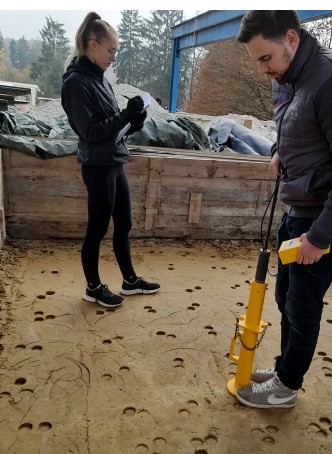

**Figure 11.** Measurements of clay subgrade on 17 October 2022 (**left**), 21 October 2022 (**middle**) and 3 November 2022 (**right**) by Clegg and LDD 100.

## 5. Measurement Results of Clay Subgrade Characteristics $E_{vd}$ and CIV

The measurements were carried out on 17 October 2022, 21 October 2022, and 3 November 2022, and the soil moisture content was monitored during each measurement. The results in the form of 3D plots of the measured $E_{vd}$ and CIV characteristics are shown in Figures 12–14.

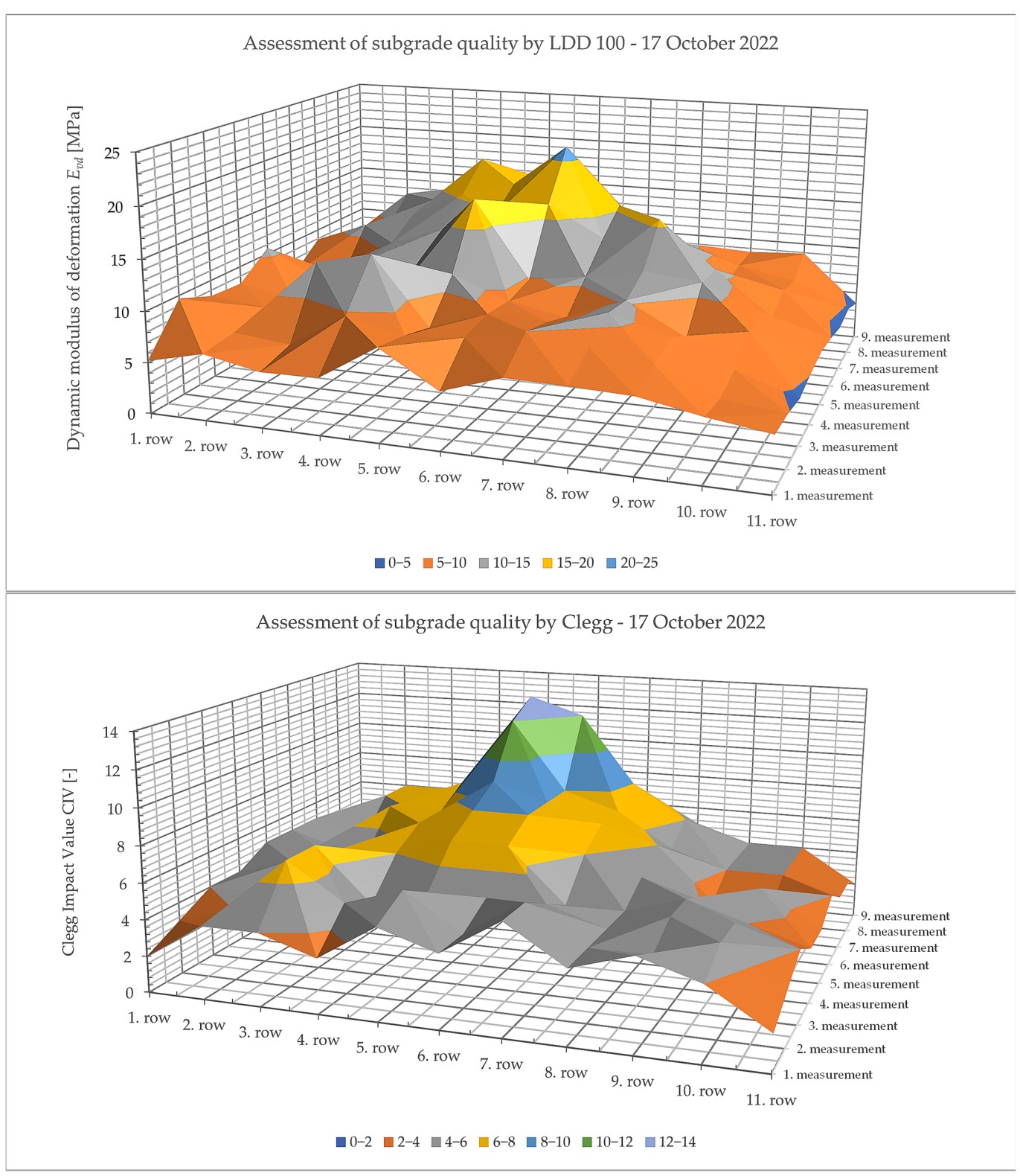

**Figure 12.** Measured values of $E_{vd}$ [MPa] and CIV of the clay subgrade at moisture content $w$ = 19.08%.

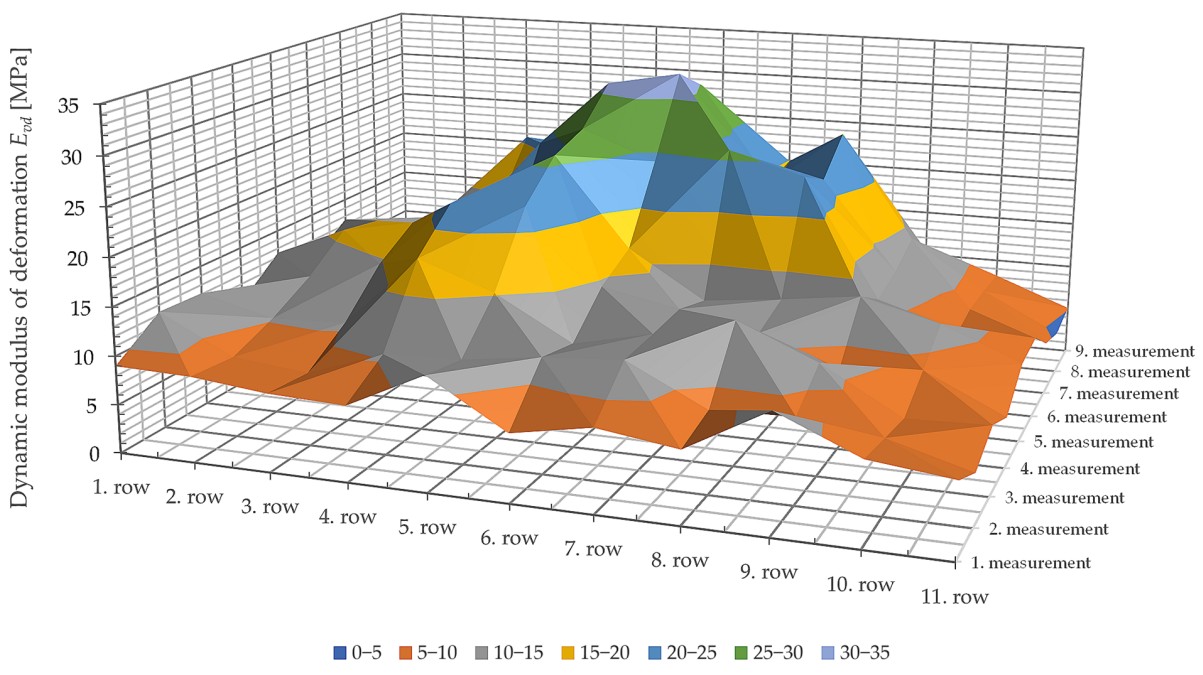

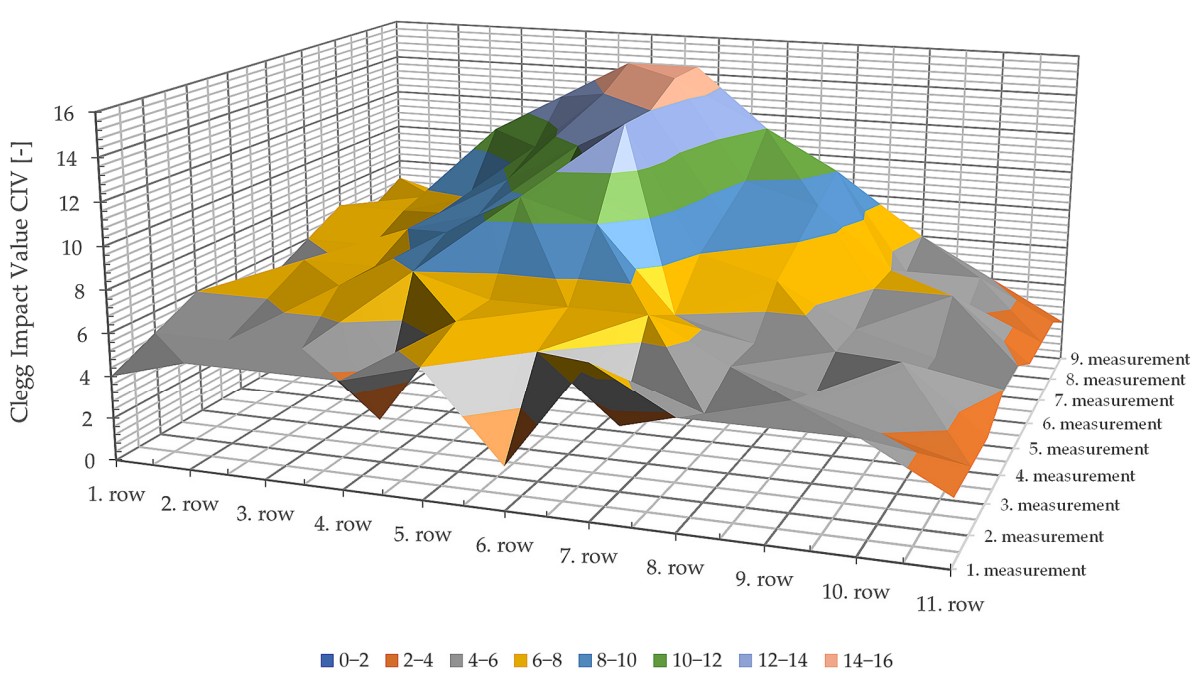

**Figure 13.** Measured values of $E_{vd}$ [MPa] and CIV of clay subgrade at moisture content $w$ = 18.63%.

Based on the measured data, linear correlations of the dependence of the impact modulus of deformation on the CIV value were observed. From the set of data, the following values were excluded by agreement:

- When the values were significantly affected by significant moisture (visual assessment);
- When the CIV value exceeded the $E_{vd}$ value, which is outside the scope of all correlations;

- When the $E_{vd}$ value exceeded the CIV value by more than 2.5-fold, which is outside the range of all correlations.

  From the modified set of data, the correlations of interest were objectified. Figures 15–17 present them in graphical form.

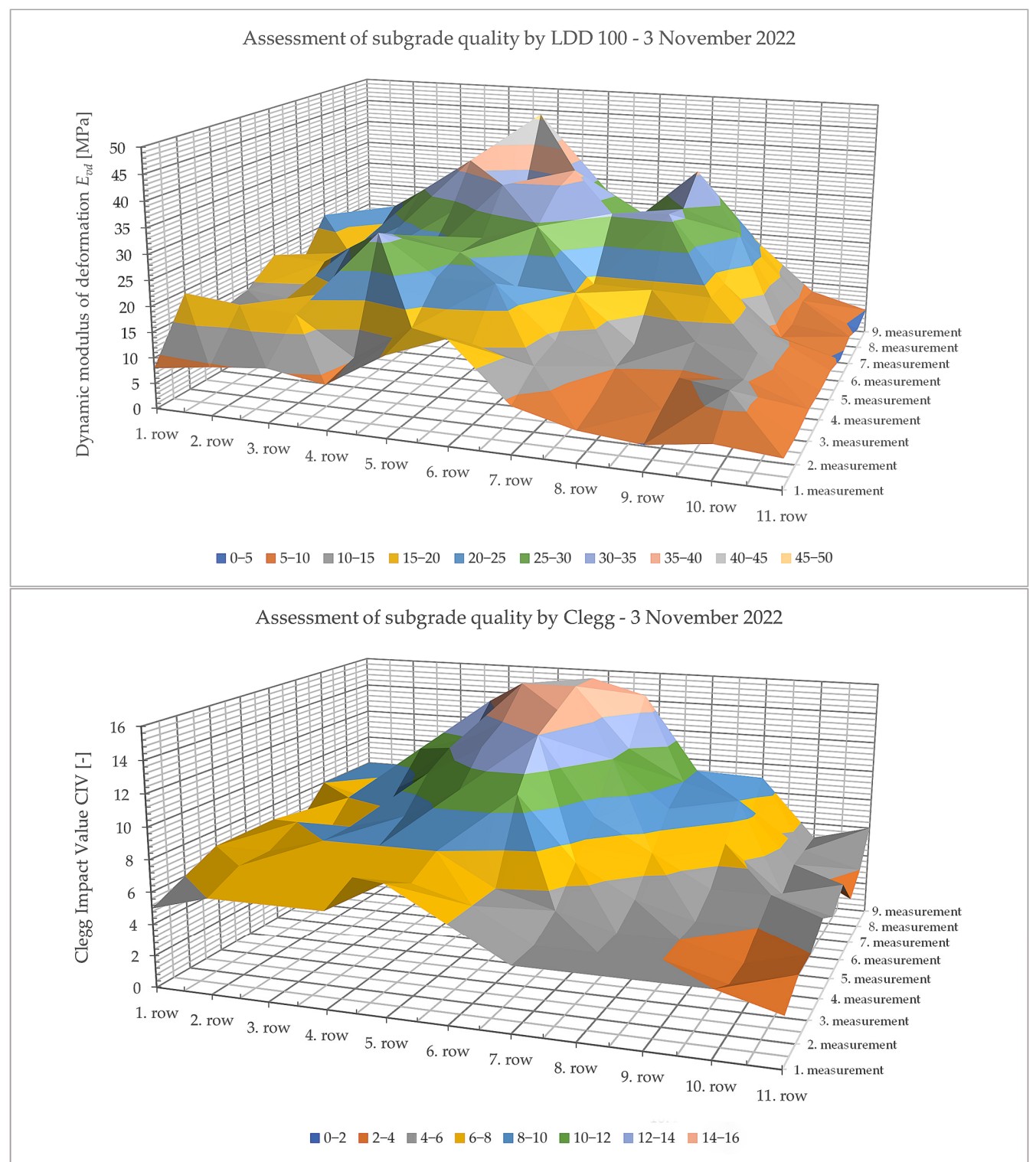

**Figure 14.** Measured values of $E_{vd}$ [MPa] and CIV of the clay subgrade at moisture content $w = 17.54\%$.

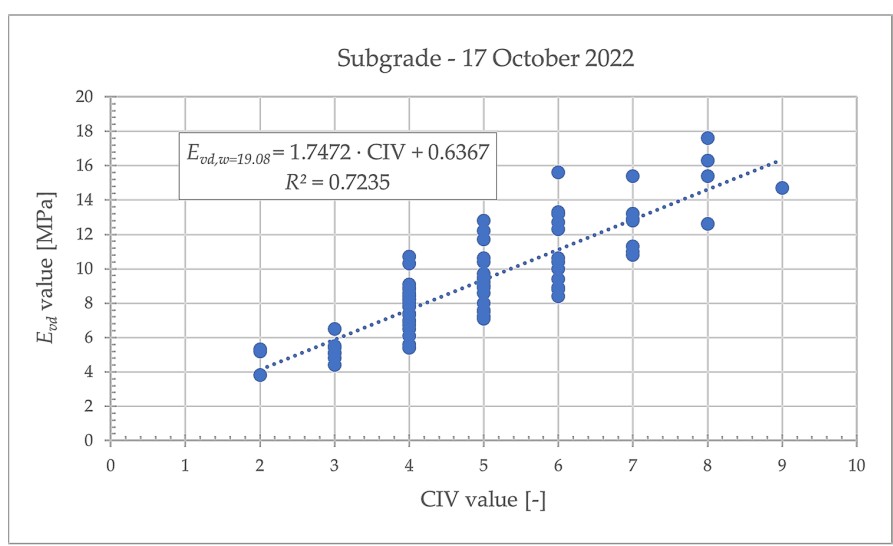

**Figure 15.** Correlation dependence of $E_{vd}$ on CIV—clay subgrade 17 October 2022.

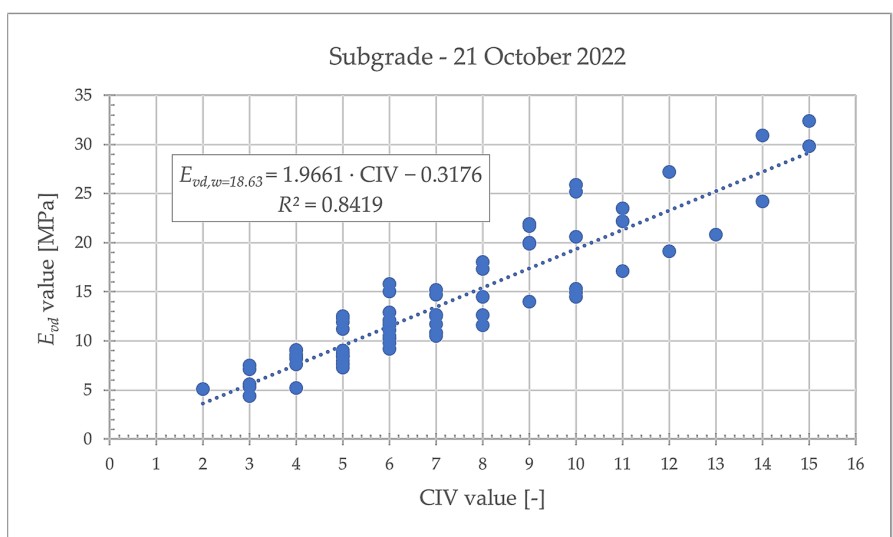

**Figure 16.** Correlation dependence of $E_{vd}$ on CIV—clay subgrade 21 October 2022.

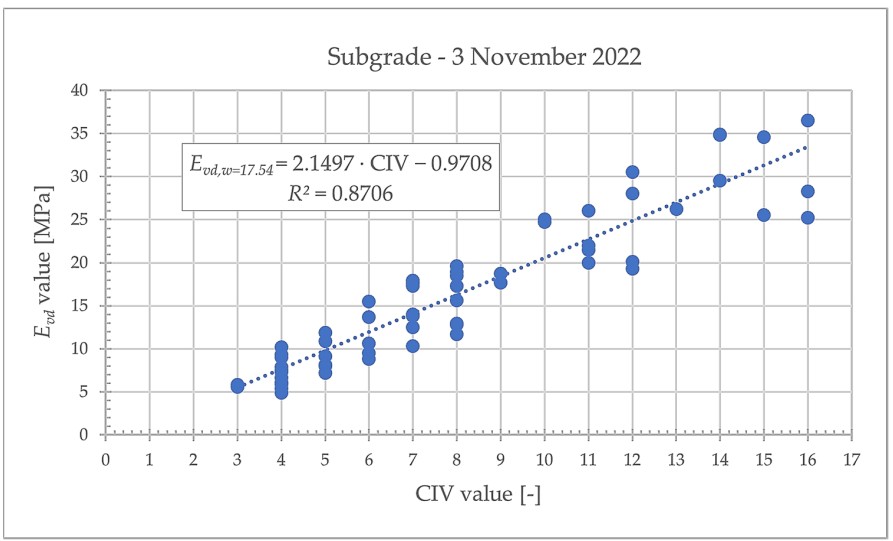

**Figure 17.** Correlation dependence of $E_{vd}$ on CIV—clay subgrade 3 November 2022.

## 6. Discussion

We further consider the most important outcomes of this research to be the following:

- As part of the verification of the suitability of the CIST device, its ability to evaluate the deformation modulus of fine-grained soils even at compressions greater than 20 mm, which is the recommended value according to the original 1980 manual for the device, was identified. The high value of the correlation (Figure 5) of the CIST hammer compression with the Clegg Impact Value (CIV) $R = 0.9890$ (very strong correlation—Table 1) induces the assumption that the CIST can be used to assess the quality of cohesive soils up to a compression value of 40 mm, which corresponds to a CBR of 2.2% in the case of homogeneous cohesive soils. This fact allows its applicability for subgrades with a bearing capacity characterized by a modulus of deformation from 20 MPa, which is the lower value recommended by the authors for non-motorized roads;
- Objectification of Formula (4), allowing for the back-calculation of CIV [-] from the measured value of $y_{CIST}$ [mm];

$$CIV = -\frac{ln(y_{CIST}/53.717)}{0.148}$$

- The dependence of $CBR_{Clegg}$ [%] on the moisture of clay soils ranged from 5 to 19% (Figure 6) according to the following relationship: the correlation showed a very strong correlation (Table 1);
- Converting CBR values evaluated from CIST measurements to CBR values evaluated according to STN 72 1016 [13];

$$CBR_{STN72\ 1016} = 0.44{\cdot}CBR_{Clegg} + 2.53 \tag{8}$$

- In general, the Clegg methodology for CBR determination shows significantly higher values (Figure 10) than the authors' observations and the results of foreign authors [12,20–23];
- The average $CBR_{Clegg}/CBR_{STN721016}$ ratio was found to be 1.8-fold.

The detailed measurements of the bearing capacity of the clay subsoil at the Scientific Research Workplace were carried out using the LDD 100 and CIST devices. The detailed results of the measurements are presented in the 3D plots in Figures 12–14. The figures demonstrate that the isomorphic model of the natural bedrock with dimensions of 3.1 by 3.8 m was considerably inhomogeneous concerning the evaluated parameters. In total, 99 measurements were performed using the LDD 100 device and the CIST at the same measurement points for each series of measurements. Based on the observed values (Figures 15–17), the following correlations of the dynamic modulus of deformation $E_{vd}$ with the Clegg Impact Value (CIV) were determined for different clay subsoil moisture $w$.

$$E_{vd,w=17.54} = 2.15{\cdot}\text{CIV} - 0.97 \tag{9}$$

$$E_{vd,w=18.63} = 1.97{\cdot}\text{CIV} - 0.32 \tag{10}$$

$$E_{vd,w=19.08} = 1.75{\cdot}\text{CIV} + 0.64 \tag{11}$$

All correlations from (8) to (11) demonstrated very strong correlations (Table 1). For the credibility comparison of the linear correlation dependence directions, Formulas (9)–(11) were transformed into a normalized form, i.e., the correlation dependencies passing through the number 0 (Figure 18).

When assessing the quality of the pavement subbase, as with other construction components of the transport infrastructure, it is necessary to apply a systematic approach considering legitimate economic and environmental aspects. It is also the requirements of the users, the inhabitants of the road environment, and the economic level of society. The significant output of the authors for the systematic approach of interest was the presentation of the credibility correlation dependence of $E_{def,2}$ determined by the static plate load test according to STN 73 6190 from the modulus of deformation $E_{vd}$ measured by LDD 100 (Figure 19) [24].

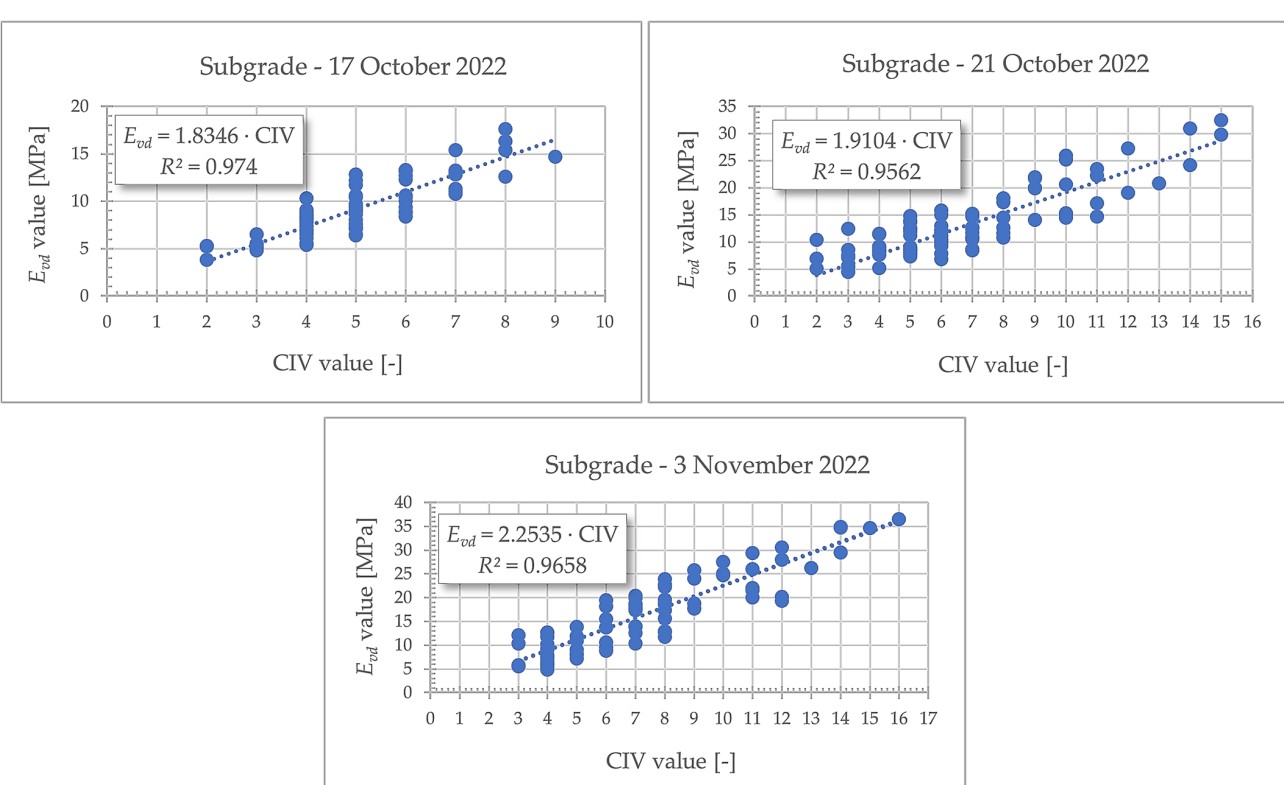

**Figure 18.** Correlation dependencies passing through the number 0.

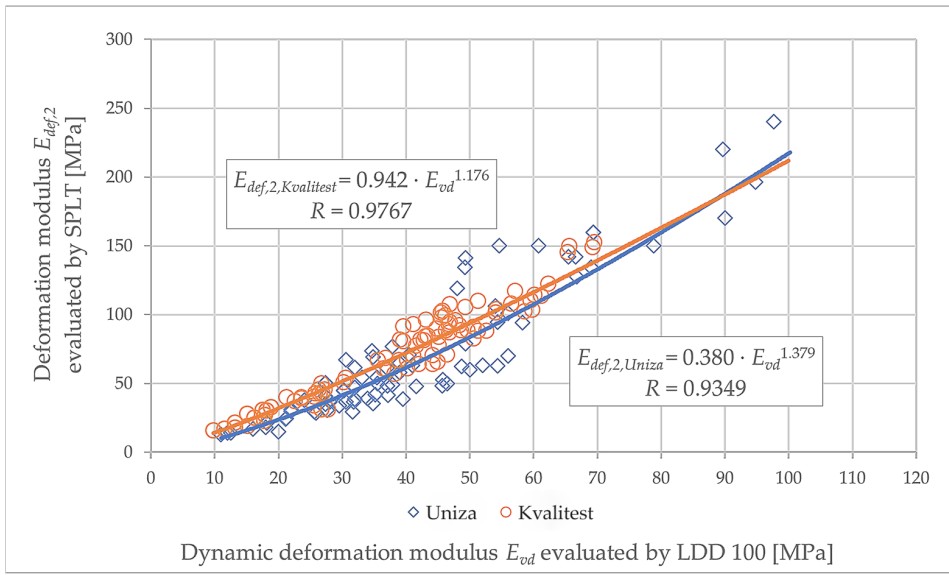

**Figure 19.** Power and linear dependence of the modulus of deformation from the second loading cycle $E_{def,2}$ on the design value of the modulus of elasticity according to [24].

## 7. Conclusions

This article represents the case study of the long-term research of the authors [12], enabling the use of the innovative CIST (Clegg Impact Soil Tester) device for quality control of fine-grained soils and bulk materials in Slovakia. This device can be mainly used where the static plate load test of building constructions [7,24–26] or the plate bearing test of pavements and subgrades by FWD [8,24] is not possible due to spatial, temporal, or financial reasons. Homomorphic models of clay subgrades (Figure 1) and isomorphic

models of pavements with clay and clay subgrades (Figures 2 and 3) have been constructed in almost 20 years of research activities.

We consider the most important results of our long-term research to be as follows:

- Identification of the ability of CIST to assess the quality of cohesive soils up to a compression value of 40 mm, corresponding to a CBR of 2.2% and a modulus of subgrade deformation of 20 MPa;
- Objectification of Formula (4), which allows the back-calculation of CIV [-] from the measured value of $y_{CIST}$ [mm];
- Finding a very strong correlation dependence of $CBR_{Clegg}$ [%] on the moisture content of clayey soils ranging from 5 to 19%.

Very detailed measurements of the bearing capacity of the clay subsoil in the Scientific Research Workplace of the Faculty of Civil Engineering of the University of Zilina were carried out using the LDD 100 and CIST instruments. The detailed results of 99 measurements with LDD 100 and CIST at the same site for each series of measurements are presented in 3D plots in Figures 12–14. Based on these in situ values, correlations of the dynamic modulus of deformation $E_{vd}$ with the Clegg Impact Value (CIV) for the clay subgrade moisture content $w$ ranging from 17.54 to 19.08 were objectified in the laboratory results (Figure 6).

Based on the objectified research results, the authors foresee the application of CIST for the systematic quality control (Figure 20) of the subsoil, including its modeling [27]. These are mainly clay subsoils, consolidations of weak soils, geogrid-stabilized soils, the performance of unsterilized and triaxially geogrid-stabilized sandy soils [28–30], and Blast-Furnace Slag [31]. Generally, CIST can be used for the quality control of fine-grained soils in the context of the current upgrading of the Slovak main railway lines, which are part of the significant European corridors AGC, AGTC, and TEN-T [25,26].

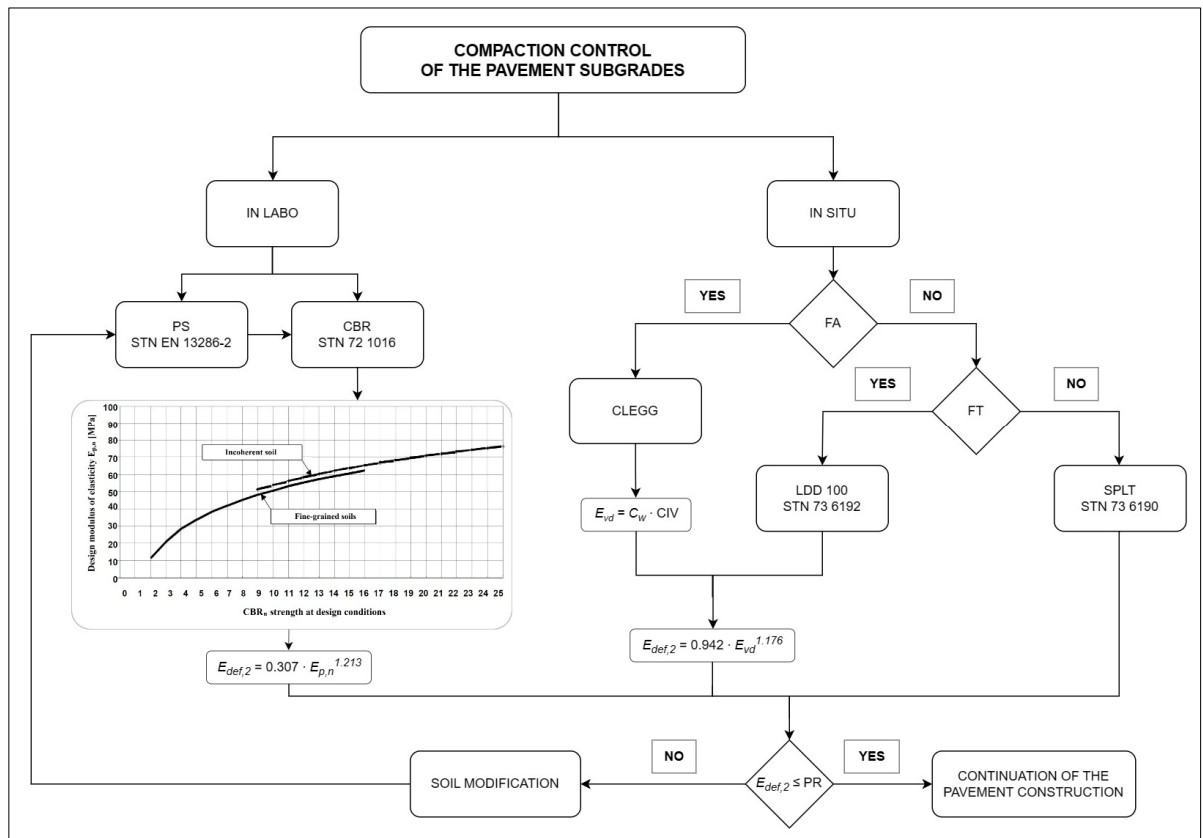

**Figure 20.** A systematic approach for compaction control of the pavement subgrades by using CIST. PS—proctor standard; CBR—California bearing ratio; SPLT—static plate load test; LDD 100—lightweight

dynamic plate; FA—area factor; FT—time factor; PR—project requirements; $E_{vd}$—dynamic deformation modulus; $C_w$—moisture coefficient (experimentally determined value as a function of moisture content); CIV—Clegg Impact Value; $E_{def,2}$—modulus of deformation from the second loading cycle; $E_{p,n}$—modulus of elasticity of the subgrade.

The decisive criteria for in situ measurements are the area factor FA (the possibility of using larger measuring devices) and the time factor TF (the time required to carry out the measurements).

**Author Contributions:** Conceptualization, M.D., J.M. and K.H.; methodology, M.D.; software, J.M.; validation, M.K.; formal analysis, K.H.; investigation, J.M. and K.H.; resources, M.D. and M.K.; data curation, J.M. and K.H.; writing—original draft preparation, M.D. and M.K.; writing—review and editing, M.D. and K.H.; visualization, K.H.; supervision, M.D. All authors have read and agreed to the published version of the manuscript.

**Funding:** This research was funded by the Ministry of Education, Science, Research and Sport of the Slovak Republic, grant number KEGA:027ŽU-4/2022.

**Institutional Review Board Statement:** Not applicable.

**Informed Consent Statement:** Not applicable.

**Data Availability Statement:** The data presented in this study are available on request from the corresponding author. At the time the project was carried out, there was no obligation to make the data publicly available.

**Conflicts of Interest:** The authors declare no conflicts of interest.

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
