# Peer review of "Innovative Quality Assessment of Pavement Subgrades Using the Glegg Impact Soil Tester"

_applsci, doi:10.3390/app14020876_

Round 1

Reviewer 1 Report

Comments and Suggestions for Authors

The work is relatively interesting, and the structure and language are well prepared. Some comments below need to be addressed or considered to improve the technical depth of the manuscript:

1. The abstract describes a lot of common sense but lacks a summary of the important conclusions of this article.

2. How was the data in Figures 5-7 obtained? And the experiment lacks experimental plans and test parameters.

3. In lines 144-179, this section describes too much common sense, it is recommended to simplify it.

4. In lines 280-350, discussion and conclusion should be listed separately.

Comments on the Quality of English Language

Minor editing of English language required

Reviewer 2 Report

Comments and Suggestions for Authors

This is interesting and timely work. Some comments:

1.       The title is too long and not easy to understand, please shorten it without losing the information.

2.       After reading this paper, I believe it is a case study work, please highlight it.

3.       Abstract: more information about CIST should be given since not everyone knows it.

4.       Introduction: the description of Horizon and the first four paragraphs is too general, I donot catch the connection between Horizon 2021-27 and the present study. Please be more specific.

5.       Figures 12-14 are perfect.

6.       Figure 20: please convert this figure into a flow chart with judgment.

Comments on the Quality of English Language

OK for me.

Reviewer 3 Report

Comments and Suggestions for Authors

Comments on the Quality of English Language

The writing is readable.

Reviewer 4 Report

Comments and Suggestions for Authors

While the study aims to advance an innovative soil testing method, some aspects could be strengthened:

1.         The description of testing protocols and derivation of equations lacks details, making it difficult to fully replicate the results and thus threatening reliability. More information on procedures is needed.

2.         Comparisons are absent between experiment outcomes under different years/conditions, without analyzing potential influences from external factors. This would strengthen conclusions.

3.         The paragraph from line 84 to 93 lacks crucial information regarding the measurement procedure of the CIST device and potential factors that could introduce bias in the measurement data.

4.         Table 1's qualitative description of correlation levels is too vague for quantitative judgments. More specific classification criteria referenced from literature could be provided.

5.         Line 119 should include the basic physical parameters of the measurement object to provide a more comprehensive understanding of the experimental setup and the variables at play.

6.         Line 196 should explicitly state the relationship between the CBR value and soil moisture content, as this is essential for interpreting the results and drawing accurate conclusions.

7.         Line 240: The discrepancy in the data needs to be addressed and explained in order to ensure the integrity of the study’s findings. Without a clear explanation for this discrepancy, the reliability of the results is called into question.

8.         The limited soil sample types and sizes may not sufficiently represent real engineering scenarios due to constraints. Larger and more diverse samples would increase persuasiveness.

9.         Correlations are primarily linear regressions, neglecting other possible complex relationships. Comparison of multiple models' predictive performance would be insightful.

10.     Directly applying cist test results in design ignores possible impacts from inherent errors. Further validation of its applicability in engineering practice is needed.

11.     Many conclusions lack numeric support, such as quantifying the increased applicable soil types for cist.

12.     Some statements could be improved in terms of expression fluidity and detail precision.

Overall, the research value is undoubted but deeper analysis and refinement on specifics leave room for enhancement. I hope these comments provide constructive input to elevate the manuscript quality.

Round 2

Reviewer 3 Report

Comments and Suggestions for Authors

suggest to be accepted as is